# HER2 Oncogene as Molecular Target in Uterine Serous Carcinoma and Uterine Carcinosarcoma

**DOI:** 10.3390/cancers15164085

**Published:** 2023-08-14

**Authors:** Blair McNamara, Levent Mutlu, Michelle Greenman, Justin Harold, Alessandro Santin

**Affiliations:** Department of Obstetrics, Gynecology, and Reproductive Sciences, Yale University School of Medicine, 333 Cedar Street, LSOG 305, P.O. Box 208063, New Haven, CT 06520, USA

**Keywords:** HER2, uterine serous carcinoma, carcinosarcoma, antibody–drug conjugate, gynecologic malignancy, chemotherapy, endometrial cancer

## Abstract

**Simple Summary:**

There are several types of uterine cancer. Two of the more rare types are uterine serous carcinoma and carcinosarcoma, which are both very aggressive forms. These rare, aggressive types often have mutations that allow them to be treated with specific therapies that are targeted. One such mutation causes an increase in proteins named HER2 on the tumor cell surfaces. This article summarizes the evidence behind all of the different HER2-targeting therapies for uterine serous carcinoma and uterine carcinosarcoma.

**Abstract:**

Uterine serous carcinoma (USC) and uterine carcinosarcoma (UCS) are two rare histologic variants of uterine carcinoma, with distinct molecular profiles and aggressive metastatic potential. As the effectivity of traditional platinum-based chemotherapy for USC and UCS is low, and there are high rates of resistance and recurrence, the development of novel targeted therapeutics is needed. Human epidermal growth factor receptor 2 (HER2) has proven to be an oncogene of increasing interest in these cancers, as HER2 protein overexpression and/or c-*ERBB2* gene amplification ranges from ~30 to 35% in USC, and between ~15 and 20% in UCS. This review summarizes the existing clinical and preclinical evidence, as well as ongoing clinical trials of HER2-targeting therapeutics, and identifies potential areas of further development and inquiry.

## 1. Introduction

Cancer of the endometrium is the most common gynecologic cancer in developed countries. In the United States in 2022, it is estimated that there were 65,950 new diagnoses and 12,550 deaths attributed to endometrial cancer [1]. The incidence of endometrial cancer is increasing each year and is expected to surpass colorectal cancer as the third most common cancer by 2040 [2]. Uterine serous carcinoma (USC) is a high-grade histologic subtype of endometrial cancer, with an aggressive biology and high rates of recurrence and low survival. For diseases at later stages (i.e., stage III–IV), 5-year survival is estimated to be only 37% [3]. Despite accounting for only up to 10% of endometrial cancers, USC is responsible for up to 40% of endometrial cancer-associated deaths [4]. Uterine carcinosarcomas (UCS), previously known as Mixed Müllerian tumors (MMMT), are also a rare and biologically aggressive subtype of endometrial cancer. UCS accounts for less than 6% of all gynecologic cancers [1,2]. Five-year survival rates of UCS are significantly lower than that of endometrial carcinoma [3,4,5]. The biologic profile of USC and UCS, their rarity, resistance to standard chemotherapy, as well as their outsized mortality underscore the need for development of novel more effective targeted therapies, especially for advanced and recurrent disease.

Human epidermal growth factor receptor-2 (HER2) is encoded by the *ERBB2* gene, which when amplified, causes HER2 to be overexpressed in many different solid malignancies. HER2 is a member of the human epidermal growth factor receptor (HER) family of tyrosine kinase receptors. When activated, it dimerizes with other tyrosine kinase receptors in the HER family (HER1/EGFR, HER3, and HER4) and induces signal transduction through the mitogen-activated protein kinase and PI3K signaling cascade [5]. This tyrosine kinase activity leads to the induction of genes that further promote malignant cellular potential. In gynecologic malignancies, HER2 is most often overexpressed in endometrial cancer (17–30%) and ovarian cancer (5–60%) [6,7,8]. Specifically, HER2 expression in USC is estimated to be 30–35% [9,10], while rates of HER2 expression in UCS in the literature are more varied, estimated to be between 6% and 56% [2,10]. Like in many cancers, HER2 has been demonstrated to be a poor prognostic marker, but also represents a potential target for novel therapeutics [7,11]. In this review, we discuss how the HER2 receptor functions in USC and UCS, evidence supporting various detection and quantification methods, and contemporary therapeutic updates that are drastically changing treatment paradigms for patients with HER2-positive USC and UCS. 

## 2. HER2 Function and Role in Tumorigenesis

The human HER2 (c-erbB2) gene product encodes a receptor protein that includes a cysteine-rich extracellular ligand binding domain, a hydrophobic transmembrane region, and an intracellular tyrosine kinase domain [12]. HER-2/neu functions as a preferred partner for heterodimerization with other members of the EGFR family (ErbB1, ErbB3, and ErbB4), and therefore plays a major role in coordinating the complex ErbB signaling network that is responsible for regulating cell growth and differentiation [12]. HER2 gene amplifications and mutations have been identified in a variety of cancers [13].

## 3. Considerations for HER2 Testing and Evaluation

Tumor testing of HER2 expression can be performed through various methods, and no standardized method nor scoring system exists for testing USC and UCS tumors. Testing options that exist include the immunohistochemistry (IHC) assay, which is scored based on a percentage of positive tumor cells and the intensity of cell membrane staining. Fluorescence in situ hybridization (FISH) and next-generation sequencing (NGS) can also be used. 

### 3.1. Testing Method Discordance

Recent studies have sought to evaluate whether rates of HER2 positivity differed with the use of various testing methods and scoring systems in USC. The Gynecologic Oncology Group (GOG) published data from a large cohort of patients with advanced or recurrent endometrial carcinomas; among the patients with USC tumors, 61% were found to have HER2 overexpression by IHC staining, and only 21% were found to have HER2 gene amplification by FISH [14]. In 2013, Buza et al. reported that among USC samples, IHC and FISH concordance was 75–81% depending on which scoring criteria were used [15]; as these both fell below the recommended concordance of 95% described in ASCO/CAP guidelines for HER2 scoring of breast cancer, the authors argued for standardized IHC and FISH testing on all USC cases until testing concordance improves [16]. In their 2022 study, Klc et al. evaluated 2192 USCs, and reported that rates of HER2 positivity in their cohort varied between 12.3 and 16.3% using different IHC criteria, 18.6–19.6% using different ISH criteria, and 10.5% by NGS [17]. 

A recent survey study reported that in the United States, laboratories use discordant practices for HER2 testing in USC. The most common is IHC, with reflex to ISH for equivocal results (254 of 286, 88.8%). Most reported scoring criteria were 2018 ASCO/CAP guidelines (195 of 281 (69.4%)). Only 16.0% (45/281) reported using guidelines specific to USC [18], based on the Fader et al. 2018 clinical trial guidelines [19]. These data raise the concern that non-standardized testing methods across laboratories could significantly impact patients’ qualification for HER2-directed therapy; a further concern is that different testing methods have not yet been correlated to the response to HER2-directed therapy. 

### 3.2. HER2 Intratumoral Heterogeneity

Further complicating HER2 classification is that intratumoral heterogeneity of HER2 expression (defined as at least 5% of tumor having at least a 2-degree change in HER2 expression intensity [20]) is quite common in USC, ranging between 31% and 100% of cases [21].

### 3.3. Novel Testing Methods

At present, NGS technology is being evaluated for its potential utility of assessing HER2 tumoral status. In USC, few studies exist; however, those that do report NGS as highly specific for HER2 gene amplification, but less sensitive than IHC or FISH in assessing tumors with focal expression or low levels. Klc et al. published their large cohort study with concordance rate of 91.6% and positive predictive value of 60.3% using NGS testing to detect HER2 amplification in USC compared with chromogenic ISH results [17].

## 4. HER2 as Prognostic Marker

There is significant literature at this point that supports HER2 as a poor prognostic indicator in USC. In 2005, Santin et al. reported overexpression of HER2 as independently associated with poor survival outcomes in a cohort study of 27 patients [22]. In their multi-institutional cohort study of 169 patients with stage I USC, Erickson et al. reported a 26% HER2-positivity rate. With no significant differences between HER2-positive and negative cohorts in terms of age, stage, adjuvant therapy, and follow-up, HER2-positive patients had increased lymphovascular space invasion, more recurrences, and worse PFS and OS. After controlling for potential confounders on multivariate analysis, patients with HER2 positive tumors experienced worse PFS, with a hazard ratio (HR) of 3.5 (95% CI 1.84–6.67; *p* < 0.001) compared to patients with HER2 negative disease [23].

## 5. HER2-Targeting Therapies

### 5.1. Monoclonal Antibody Therapies

Trastuzumab is the first HER2-directed monoclonal antibody approved by the FDA in 1998 for treating HER2-positive metastatic breast cancer. Trastuzumab has tumoricidal efficacy through both antibody-dependent cellular cytotoxicity (ADCC), as well as inhibition of tumor cells’ innate HER2-mediated signaling pathway, blocking downstream effects of cell proliferation and angiogenesis [24]. It is FDA approved for use in HER2-positive early-stage and metastatic breast cancer, and HER2 positive gastric and gastroesophageal junction adenocarcinoma [25,26]. 

One major Phase II trial has evaluated the clinical efficacy and safety of trastuzumab in USC. In 2018, Fader et al. published results of a multi-center randomized phase II trial of carboplatin-paclitaxel versus carboplatin-paclitaxel-trastuzumab in advanced (stage III-IV) or recurrent USC that overexpress HER2 (NCT01367002) [19]. For this trial, patients were considered to have tumors that overexpressed HER2 if IHC score was 3+, or 2+ with *ERBB2* gene amplification confirmed by FISH testing. The trial enrolled 61 patients between August 2011 and March 2017, and had a primary endpoint of progression free survival (PFS). They reported a median PFS of 8.0 months for the control arm, versus 12.6 months (*p* = 0.005; HR 0.44; 90% CI, 0.26 to 0.76). The cohort receiving primary treatment (i.e., stage III-IV patients, *n* = 41), PFS was 9.3 vs. 17.9 months (*p* = 0.013; HER 0.40; 90% CI, 0.20 to 0.80). They reported similar toxicity between the two arms. In 2020, Fader et al. published an updated survival analysis that confirmed the initial PFS findings. They further reported overall survival (OS) differences, favoring the trastuzumab cohort. This difference was most notable in patients with stage III to IV disease (as opposed to recurrent disease), with median survival of 24.4 months in the control arm vs. not reached in the trastuzumab arm (HR 0.49; 90% CI, 0.25–0.97; *p* = 0.041). In patients with recurrent disease (*n* = 17), patients who received trastuzumab in addition to standard chemotherapy had a 3.2-month increase in PFS (HR 0.14; 90% CI, 0.04–0.53; *p* = 0.003) [9]. 

Based on these data, trastuzumab was listed in the National Comprehensive Cancer Network (NCCN) guidelines as the preferred treatment regimen when used in combination with carboplatin-paclitaxel for patients with stage III/IV or recurrent USC with HER2 overexpression (a category 2A recommendation) [9]. To date, clinical data have not been published on the utility of trastuzumab in uterine carcinosarcoma. However, preclinical studies have demonstrated in vitro and in vivo efficacy of trastuzumab in UCS [27]. 

Pertuzumab is another monoclonal antibody that binds to an extracellular domain of HER2; it has a different epitope than trastuzumab. It is FDA approved for HER2+ breast cancer in combination with trastuzumab and chemotherapy [28]. There is an ongoing phase II/III clinical trial (NCT05256225) evaluating the addition of trastuzumab or trastuzumab with pertuzumab to chemotherapy for HER2 positive, myoinvasive stage I-IV USC or UCS (Table 1). Patients with recurrent disease are not eligible for participation, and patients must be within 8 weeks of their primary surgery. Primary study completion is expected in October 2027, and expected enrollment is 525 patients. 

### 5.2. Tyrosine Kinase Inhibitors

Other small molecule HER2-directed therapies target the intracellular tyrosine kinase domain of the HER receptors. Several of these drugs exist and are FDA approved for use in other cancers; none are currently approved for use in gynecologic malignancies. Lapatinib and neratinib are both HER2 tyrosine kinase inhibitors approved for use in HER2-positive breast cancer [29]. They have not been studied clinically in gynecologic malignancies; however, there is preclinical evidence supporting the use of neratinib alone or in combination with olaparib, in HER2 overexpressing uterine serous carcinoma [30,31]. Afatinib is another small molecule oral tyrosine kinase inhibitor that showed excellent preclinical activity against HER2-amplified USC [32]; it is currently being studied in a phase II clinical trial of patients with persistent or recurrent HER2-positive USC (NCT02491099), with estimated primary completion of 2024 (Table 1). 

### 5.3. Antibody—Drug Conjugates under Clinical Evaluation

Antibody—drug conjugates (ADCs) are a novel class of targeted therapies. ADCs are composed of a humanized monoclonal antibody specific for a tumor surface antigen such as HER2, a cleavable or uncleavable linker, and a cytotoxic payload; they thus enable a more targeted delivery of tumor-directed therapy, while minimizing toxicity to normal tissues and thus fewer adverse effects than traditional chemotherapy [33]. There are not currently any ADCs that are FDA approved for use in uterine serous or uterine carcinosarcomas; however, there are several ongoing trials evaluating their efficacy and safety in USC and UCS. 

#### 5.3.1. Trastuzumab Deruxtecan

Trastuzumab deruxtecan (T-DXd, or DS-8201a) is a novel HER2-directed antibody–drug conjugate. It is composed of the humanized monoclonal antibody trastuzumab, which targets extracellular domains of HER2 cell surface receptors, a cleavable tetrapeptide linker (maleimide glycine-phenylalanine-glycine (GGFG) peptide), and a topoisomerase I inhibitor payload. This tetrapeptide linker is designed to be cleaved by lysosomal enzymes such as cathepsins B and L, which are overexpressed in tumor cells [34]. After T-DXd is internalized into tumor cells, the toxic payload, DXd, is released and can permeate through the cell membrane to neighboring tumor cells, regardless of their HER2 status; this feature has resulted in T-DXd demonstrating clinical efficacy in heterogenous HER2 tumors [35]. 

With promising preclinical data with T-DXd in HER2-amplified carcinosarcoma mouse xenograft models [36], T-DXd has now been evaluated in a number of clinical trials for gynecologic malignancies, some of which are ongoing. The STATICE trial is a multicenter phase II, single-arm study that took place across several centers in Japan. This trial evaluated T-DXd in patients with advanced or recurrent uterine carcinosarcomas (UCS), of various HER2 expression, previously treated with chemotherapy [37]. In this study, 32 patients were stratified based on HER2 expression: 22 with HER2-high USC and 10 with HER2-low. T-Dxd was administered at two doses (6.4 or 5.4 mg/kg) intravenously every 3 weeks until disease progression, unacceptable toxicity, or withdrawal of consent. The authors report an overall response rate (ORR) of 54.5% (95% CI, 32.2 to 75.6) in the HER2-high cohort, and 70% (95%CI 45.1 to 86.1) in the HER2-low group. They report a median duration of response (DoR) of 6.9 months in HER2-high group and 8.1 months in HER2-low group. Both groups demonstrated a 100% disease control rate. 

In the United States, a Phase I trial (NCT02564900) administered T-DXd to patients with HER2-positive solid tumors. This cohort included 59 patients, 2 of which had endometrial cancer, and 1 patient with cervical cancer. The reported median DoR was 11.5 (95% CI 7.0 to not met) months, and median PFS was 7.2 months (95% CI 4.8 to 11.1). Patients with HER2-expressing tumors had the most pronounced tumor shrinkage [38]. 

The subsequent Phase II, multicenter, open label trial (DESTINY-PanTumor02, NCT04482309) (Table 1), has completed enrollment of 267 participants with bladder, biliary tract, cervical, endometrial, ovarian, and pancreatic cancer that express HER2. Primary outcome measures were ORR, and secondary outcome measures were DoR, PFS, OS, and safety data. The study is closed to enrollment and is continuing to collect outcome data. Encouraging preliminary data were reported at the ASCO 2023 annual meeting; in the uterine cancer cohort, 57.7% of all patients responded to treatment (i.e., complete or partial response). Of patients with IHC 3+ HER2 tumors, 84.6% responded to treatment, and of those with IHC 2+ HER2 tumors, 47.1% responded. T-DXd has shown promising preclinical activity in other gynecologic malignancies, such as ovarian carcinosarcoma [36]. T-DXd has received FDA approval for use in HER2-positive or HER2-low metastatic breast cancer, HER2 positive gastric cancer, and HER2 mutated non-small cell lung cancer [39,40,41]. A recently published case report described a patient with recurrent, metastatic, treatment-resistant USC with HER2 overexpression and c-*ERBB2* amplification. The patient was treated with T-DXd and experienced a significant reduction in disease burden, disappearance of metastatic back bone pain, as well as normalization of elevated CA-125 quickly after starting treatment [42]. 

#### 5.3.2. Trastuzumab Duocarmazine

Trastuzumab duocarmazine (SYD985) is composed of trastuzumab (a HER2-directed monoclonal antibody) covalently bound to a duocarmycin derivative. This payload is attached via cleavable linker and contains the prodrug *seco*-duocarmycin-hydroxybenzamide-azaindole (*seco*-DUBA). DUBA effects its tumoricidal properties by alkylating DNA, resulting in DNA damage and cell death. There is one completed phase I trial and an ongoing phase II trial available evaluating this novel ADC in USC and UCS.

In a Phase I dose-expansion clinical trial of trastuzumab duocarmazine in patients with HER2-expressing breast, gastric, urothelial, or endometrial cancer, 146 patients were enrolled, 14 of them with endometrial cancer. Patients were given 1.2 mg/kg of trastuzumab duocarmazine every 3 weeks. Of the 14 patients with endometrial cancer, 5 (39%, 95% CI 13.9 to 68.4) patients had partial disease responses [43]. 

A Phase II open-label, single-arm study of trastuzumab duocarmazine in patients with HER2-expressing recurrent, advanced, or metastatic endometrial carcinoma (Table 1) completed its initial enrollment and data collection on 25 April 2023 (NCT04205630). Investigators enrolled 64 patients who experienced disease progression on or after first line platinum-based chemotherapy. Patients with two or more prior lines of chemotherapy for advanced/metastatic disease were not eligible. Primary outcomes include ORR, PFS, OS, and treatment-emergent adverse events (AEs). HER2 immunohistochemistry (IHC) scores of 1+, 2+, and 3+ were allowed. Results from this trial have not yet been released. 

Trastutzumab duocarmazine has shown preclinical antitumor activity in HER2-expressing epithelial ovarian carcinoma, uterine serous carcinoma, and uterine and ovarian carcinosarcomas [44,45,46].

### 5.4. Antibody—Drug Conjugates under Preclinical Evaluation

Two ADCs, trastuzumab emtansine (T-DM1) and DHES0815, both have excellent preclinical data supporting their use in USC and UCS; however, neither are currently being evaluated in clinical trials. 

In a study of HER2-overexpressing xenograft UCS models, T-DM1-treated mice experienced slower tumor progression in vivo compared to trastuzumab treatment alone. An in vitro component also demonstrated inhibition of UCS cancer cell proliferation with T-DM1 treatment [47]. One case report exists of a patient with recurrent, metastatic, treatment-resistant USC with HER2 overexpression (3+ by IHC) who received TDM-1 and experienced complete resolution of disease with prolonged systemic control [48].

DHES0815A is another ADC that utilizes a different HER2 specific humanized monoclonal antibody than trastuzumab covalently linked to a unique DNA monoalkylating agent, pyrrolo[2,1-*c*] [1,4] benzodiazepine monoamide. In preclinical studies of its activity in vitro and in vivo against primary USC cell lines and mouse xenograft models, DHES0815A had significant antitumor activity. In experiments against tumor cell lines, DHES0815A had significantly more cytotoxicity compared to a nontargeted control ADC (*p* < 0.05). USC without HER2 expression did not have significant cell cytotoxicity when exposed to DHES0815A. In mouse xenograft USC models, DHES0815A slowed tumor growth and conferred increased survival compared to control treated mice (*p* < 0.01) [49].

## 6. Resistance Mechanisms to HER2-Targeted Therapies

Tumoral resistance to HER2-directed therapy has been reported in the clinical setting, limiting the long-term effectiveness of trastuzumab therapy. Acquired or innate resistance to trastuzumab has been explained by several potential mechanisms [20,35,50,51]. The first is tumoral shedding of the extracellular domain of the HER2/neu receptor into circulation, thereby making the unconjugated antibody less effective in binding to the tumor tissue. The second is acquired PI3K-activating mutations causing downstream resistance to the induction of apoptosis in c-erbB2 amplified tumors. The third is, as described above, the heterogenous expression of the HER2/neu receptor in USC. With a tumor harboring both 3+ and 0/1+ HER2 tumor cells, HER2-directed therapy may not work up front or may diminish as HER2-negative tumor cells outgrow HER2-positive tumor cells due to trastuzumab’s selective pressure in HER2 heterogenous tumors over time. 

Understanding these potential resistance mechanisms can lead to the development of novel drug combinations as well as the development of more effective anti-HER2 therapy in USC. Many ADCs, due to their direct cytotoxic effects and bystander effects, have innate properties that can overcome some of the aforementioned resistance mechanisms, such as PI3K-activating mutations as well as heterogenous HER2 expression. In preclinical studies, combining PIK3CA inhibitors such as taselisib with CDK2/9 kinase inhibitors (i.e., CYC065), with irreversible pan-HER inhibitors (such as afatinib and neratinib) have shown promise against trastuzumab-resistant USC cell lines [52].

## 7. Conclusions

HER2-directed therapy remains of great promise in both uterine serous and uterine carcinosarcoma management. These rare and biologically aggressive gynecologic malignancies have high rates of recurrence even in patients diagnosed at early stages (i.e., stage I-II) and despite the use of adjuvant chemotherapy. There remains an opportunity for improvement and a need for novel, more effective therapies. HER2-directed ADCs show substantial promise and stand to provide significant improvements in effectivity and tolerability of cytotoxic therapy. With many phase II trials underway evaluating HER2-directed therapies in USC and UCS, the treatment of these cancers is at an exciting horizon. 

### Future Directions

Apart from novel drug development and testing in clinical trials, we believe several additional considerations regarding HER2-targeted therapies in USC and UCS should be highlighted. Firstly, as described here, the heterogeneity in HER2 scoring systems of gynecologic tumors will be a barrier to the evaluation and application of novel drugs. As more therapies come to market and are evaluated in larger clinical trials, we will need a universal HER2 scoring system, validated in clinical trials, to determine which of our patients stand to benefit most from these therapies. Secondly, preclinical evidence and preliminary clinical studies have recently shown the effectiveness of HER2-directed ADCs against HER2-low expressing USC and UCS tumors. As HER2-low breast cancer has shown promising clinical benefits from T-DXd [53], this is an exciting further area of potential benefit. Third, as more and more highly potent and targeted HER2-directed ADCs appear on the market (as well as other ADCs), we will need to evaluate their potential for combination with standard chemotherapies, as well as the potential utility in an adjuvant, first line, setting. Fourth, novel treatments for HER2 USC and UCS will need to be studied and interpreted with new guidelines based on the TCGA molecular classifications. The vast majority of USC will fall into the p53 aberrant group; UCS will also mostly be classified as p53 aberrant, but may have a more heterogenous profile [54]. Unfortunately, the studies summarized here do not report on molecular classifications of the USC and UCS cancers evaluated; this could be because the studies were designed prior to publication of the TCGA classification, or due to financial or logistic considerations. While molecular classification has been shown to be a more relevant prognostic indicator than histologic subtype, the impact of the different molecular classifications on response to HER2 therapy is thus far unknown [55]. Lastly, while preliminary cost-effectiveness analysis has shown that the addition of trastuzumab with carboplatin/paclitaxel for the treatment of advanced stage/recurrent USC is a cost-effective treatment strategy [56], studies will need to be conducted evaluating the cost effectiveness of HER2-directed ADCs. Cost effectiveness studies have been performed for the ADC T-DXd against HER2 high and low metastatic breast cancer, with cautious results [57,58]. Hopefully, with increased data and applicability of these novel therapies, efforts can be made to reduce their cost and expand access to these potentially life-saving therapies.

## Figures and Tables

**Table 1 cancers-15-04085-t001:** Summary of active * clinical trials testing HER2-targeting agents in Uterine Serous Carcinoma or Uterine Carcinosarcoma.

Clinicaltrials.gov Identifiers	Phase	Drug	Drug Class	Study Population
NCT04482309	2	Trastuzumab deruxtecan (T-DXd, DS-8201a)	ADC	Locally advanced, metastatic, or unresectable endometrial cancer with HER2 overexpression (Cohort 4)
NCT04205630	2	Trastuzumab Duocarmazine (SYD985)	ADC	HER2-expressing recurrent, advanced or metastatic endometrial carcinoma
NCT04235101	1			
NCT02491099	2	Afatinib	Tyrosine Kinase Inhibitor	Persistent or recurrent HER2-positive USC
NCT05256225	2/3	Trastuzumab/pertuzumab or trastuzumab with carboplatin-paclitaxel	Monoclonal antibody	Primary, chemotherapy-naïve, HER2-positive endometrial serous carcinoma or carcinosarcoma
NCT04585958	1	T-DXd + olaparib	ADC	HER2-expressing metastatic/unresectable cancers, with dose expansion phase limited to endometrial serous carcinoma
NCT05150691	1/2	DB-1303	ADC	HER2-positive advanced solid tumor

USC, uterine serous carcinoma. * Active based on information provided on https://clinicaltrials.gov/ (accessed 30 June 2023).

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
