# Peer review of "HER2 Oncogene as Molecular Target in Uterine Serous Carcinoma and Uterine Carcinosarcoma"

_cancers, 2023, doi:10.3390/cancers15164085_

Round 1
Reviewer 1 Report
This paper gives a large view on possible treatments fo HER2 uterine serous carcinoma and carcinosarcoma. The paper organization is very clear. The only suggestion is that the efficacy of HER2-therapies on overall survival and progression free survival could be better highlighted perhaps with a table in which the studies described are included, underling the effect of different treatments on OS and PFS. Moreover, a better distinction among local and metastatic stages in this setting could be interesting, if feasible.
Author Response
We thank the reviewer for the positive comments and suggestions for this article. We would love to include more OS and PFS information, but many of the novel therapies we discuss have not had completed studies in gynecologic malignancies. There is no outcome or survival data available to summarize. The therapies included in Table 1 have ongoing trials; we would be happy to include a table of their OS/PFS results in other malignancies if the reviewer and editor believe that would add to the readability
Reviewer 2 Report
McNamara et al. have conducted a magnificent review of what has been published and written up to now, including any clinical trials still in progress, regarding targeted therapies based on the expression of the HER2 oncogene in the tissue of uterine serous carcinoma and uterine carcinosarcoma.
The review is very interesting and can be useful to anyone who treats cases of uterine neoplasms.
In my opinion, however, it would be necessary for the authors to also devote part of their discussion to the presence or absence of publications or trials under development that also consider the new classification of endometrial cancer proposed by the Cancer Genome Atlas (TCGA) and also included in the latest ESGO/ESTRO/ESP guidelines. While uterine serous carcinoma is p53abn in almost 100% of cases, this is not true for carcinosarcoma (Santoro, A et al. New Pathological and Clinical Insights in Endometrial Cancer in View of the Updated ESGO/ESTRO/ESP Guidelines. Cancers 2021, 13, 2623.)
Author Response
We thank the reviewer for their positive comments about our review. We agree that the TCGA classifications should be included in our discussion. We have edited our future directions discussion with this in mind, and have incorporated the suggested reference.
Reviewer 3 Report
I was glad to review the work of the authors regarding this very interesting article on HER2 Oncogene as molecular target in uterine serous carcinoma and uterine carcinosarcoma. Despite the major advances in gynecological oncology, there are still numerous unanswered questions regarding the role of HER2 in gynecological tumors. The manuscript is well-written and the incorporated table makes the study easy to follow.
I strongly recommend acceptance for publication of the paper after minor changes.
1) I would suggest a brief discussion on tumor markers that are used for the follow-up of this patients
2) "Tumor markers, such as CA-125 can be a useful tool that helps to distinguish between benign and malignant ovarian masses. It is now widely accepted that the tumor marker CA-125 is a predictive and prognostic factor in CA-125-positive ovarian cancers. Serum CA-125 level is a strong prognostic factor for overall survival and progression-free survival in ovarian cancer. There is an inverse relationship between serum CA-125 levels and survival in ovarian cancer. That means that a decreasing level generally indicates a positive response to cancer therapy while an increasing level indicates tumor recurrence and poor survival "
Add this information and consider citing:
https://pubmed.ncbi.nlm.nih.gov/32802022/
Is there a relationship between uterine serous carcinoma and uterine carcinosarcoma with CA 125?
Author Response
We thank the reviewers for their comments. The proposed purpose of this review is to discuss HER2 as a molecular target in uterine serous and carcinosarcoma. While discussion of CA-125 screening for uterine serous cancer surveillance is interesting and important, we are not sure if this review is the correct place for this discussion. We are also not aware of any evidence suggesting CA-125 is a prognostic biomarker in ovarian cancer; it is certainly a predictive biomarker, and that has been well studied and documented. Lastly, the citation the reviewers suggest is a case report of a woman who presented with a giant ovarian mass, which was found to be a benign lesion. It is not relevant to our discussion here and so we would request to the editors that we do not cite it. If the editors feel further inclusion of CA-125 data is warranted in this review, we would be happy to discuss this matter further.
Round 2
Reviewer 2 Report
The authors only partially responded to the observations made by the referee previously. In fact, having examined many publications and protocols of trials still in progress, it is surprising that in no case have researchers taken into consideration the fact that the response to innovative therapies could also depend on the molecular characterization of the tumors they are treating.
If this is true, the authors should discuss the reasons that may be behind this failure. Were the trials started before the molecular classification of endometrial cancer was published? Did the various investigators feel that HER2 protein overexpression was a characterization of carcinomas that in no way related to molecular classification? Other reasons McNamara et al. found to explain this important bias?
Failure to address the molecular classification of serous endometrial carcinoma and carcinosarcoma may be an even greater barrier than heterogeneity in HER2 scoring systems of gynecologic tumors in evaluating the efficacy of new drugs.
In my opinion it would be necessary for McNamara et al. to devote more space in their discussion to the reasons for the presence or absence of publications or trials under development that consider the new TCGA classification of endometrial cancer.
Author Response
We thank this reviewer for their additional comments. We agree that molecular classification of endometrial cancers should be at the forefront of future drug evaluation. While the vast majority of both uterine serous carcinomas and carcinosarcomas, the subjects of this review, will fall into the p53 aberrant molecular subtype, we agree further discussion in this review article would be beneficial to our readers. We have now expanded our discussion section to include further comments and references on this topic, and believe it has significantly strengthened this manuscript; we hope that the reviewer and editors will agree.
Round 3
Reviewer 2 Report
The additional comments with bibliographical notes inserted by the authors into the discussion effectively strengthened it. The manuscript is now acceptable for publication.